# Microbial Recycling of Bioplastics via Mixed-Culture Fermentation of Hydrolyzed Polyhydroxyalkanoates into Carboxylates

**DOI:** 10.3390/ma16072693

**Published:** 2023-03-28

**Authors:** Yong Jin, Kasper D. de Leeuw, David P. B. T. B. Strik

**Affiliations:** 1Environmental Technology, Wageningen University & Research, 6708 WG Wageningen, The Netherlands; yong.jin@wur.nl (Y.J.); kasper.deleeuw@wur.nl (K.D.d.L.); 2ChainCraft B.V., 1043 AP Amsterdam, The Netherlands

**Keywords:** PHA recycling, hydrolysis, anaerobic fermentation, PHBV, carboxylates

## Abstract

Polyhydroxyalkanoates (PHA) polymers are emerging within biobased biodegradable plastic products. To build a circular economy, effective recycling routes should be established for these and other end-of-life bioplastics. This study presents the first steps of a potential PHA recycling route by fermenting hydrolyzed PHA-based bioplastics (Tianan ENMATTM Y1000P; PHBV (poly(3-hydroxybutyrate-co-3-hydroxyvalerate)) into carboxylates acetate and butyrate. First, three different hydrolysis pretreatment methods under acid, base, and neutral pH conditions were tested. The highest 10% (from 158.8 g COD/L to 16.3 g COD/L) of hydrolysate yield was obtained with the alkaline pretreatment. After filtration to remove the remaining solid materials, 4 g COD/L of the hydrolyzed PHA was used as the substrate with the addition of microbial nutrients for mixed culture fermentation. Due to microbial conversion, 1.71 g/L acetate and 1.20 g/L butyrate were produced. An apparent complete bioconversion from intermediates such as 3-hydroxybutyrate (3-HB) and/or crotonate into carboxylates was found. The overall yields of the combined processes were calculated as 0.07 g acetate/g PHA and 0.049 g butyrate/g PHA. These produced carboxylates can theoretically be used to reproduce PHA or serve many other applications as part of the so-called carboxylate platform.

## 1. Introduction

Current fossil-based plastic usage contributes to various environmental and societal challenges such as climate change, microplastics in the environment, poor plastic recyclability, and invalid waste management, which take a great threat to the environment and our health [1]. Biobased plastics production has been increasing annually, which means a reduction in fossil resource usage [2]. Biobased biodegradable materials are manufactured from polymers such as polylactic acid (PLA), polyhydroxyalkanoates (PHA), polybutylene succinate (PBS), starch blends, etc. The PHA is commercially used for packaging, drug carriers, and agricultural films [3]. PHA polymers are microbially produced and accumulate as granules inside various living bacterial cells [4]. The PHA is extracted from the functional cell by isolation and purification. PHA polymers consist of various molecular structures such as poly(3-hydroxybutyrate) (PHB) and poly(3-hydroxybutyrate-co-3-hydroxyvalerate) (PHBV). The increased HV (hydroxyvalerate) content can decrease crystallinity as well as improve the biodegradation rate. PHBV with a low ratio (3%) of 3-hydroxyvalerate (3-HV) has similar mechanical properties to PHB [2].

Worldwide production capacities of PHA reached in total of 2.42 Mton bioplastics in 2021 and are expected to grow by 62% to 7.59 Mton of bioplastics in 2026 [5]. To foster circularity, effective recycling routes should be applied. End-of-life PHA products are not (yet) efficiently recycled at a large scale. PHA-based products do likely end up in various waste streams that are anaerobically digested, composted, or incinerated [6]. Also, PHA-based plastic littering may occur. Under natural conditions, such as the marine environment, it was shown that certain PHA films and bottles could be degraded within 0.1~0.2 years and 1.5~3.5 years, respectively [7]. PHA composting results in the release of CO_2_ which could be used for new biomass production to close the carbon cycle. However, more carbon and energy-efficient recycling routes for PHA are proposed [8]. One potential recycling route is via biogas production by anaerobic digestion of PHA [9,10], whereby biogas consequently can be used for PHA production [11,12]. The other is a direct tandem process, in which PHA can be first hydrolyzed into crotonate and/or 3-hydroxybutyrate (3-HB), after which these compounds can be directly converted into PHA by different microorganisms [13,14].

As mentioned, the PHA is well anaerobically degradable by mixed cultures such as methanogenic sludge under mesophilic as well as thermophilic conditions [15]. PHA methanogenic degradation tests revealed that hydrolysis products (such as 3-HB) and carboxylates (e.g., acetate and butyrate) are intermediates to convert PHA into biogas [16]. A recent anaerobic digestion study showed that product formation depends on the amount and type of supplied PHA. This material was provided in solid form with particle sizes of 100 to 250 µm. At low supplied PHBV or PHB concentrations (1 g/L), anaerobic sludge was able to recover biogas with ~80% carbon recovery [17]. However, at 10 or 20 g/L supplied powdered PHB and PHBV, a mixture of carboxylates (carbon recovery in the range of 10 to 18%) and biogas (carbon recovery in the range of 10% to 35%) was formed. The fermentation took 56 days and it was mentioned that a significant part of the materials remained nondegraded [17]. However, there were no data on plausible intermediates such as crotonate, 3-hydroxyalkanoates (such as 3-HB), or dimers. Interestingly, the produced carboxylates can theoretically be used to produce PHA again. To maximize the carboxylates production towards a circular PHA, methanogenesis should be hindered and hydrolysis should be improved since it is possibly a rate-limiting step.

Therefore, we propose a new route to recycle PHA-based end-of-life products (see Figure 1). After mechanical treatment to increase surface area, first, thermal alkaline hydrolysis is proposed. Consequently, the formation of carboxylates by mixed microbial species is intended. This way, PHA may be faster and more carbon efficient when recycled into carboxylates than via microbial hydrolysis. PHA, specifically PHB, has already been identified to hydrolyze into 3-hydroxybutyrate (3-HB) and crotonate by thermal alkaline pretreatment [13]. During the methanogenic degradation of PHA, 3-HB is a known intermediate and it is convertible into acetate and butyrate as known from pure-culture studies using *Ilyobacter polytropus* or *Ilyobacter delafieldii* [18,19]. Crotonate, which was not mentioned before during earlier methanogenic fermentation studies, is another key potential intermediate to produce carboxylates as well [10,16]. It is known that besides the 3-HB converting organisms *Ilyobacter polytropus* and *Ilyobacter delafieldii*, also *Clostridium kluyveri*, and *Syntrophomonas wolfei* can form acetate and butyrate from crotonate [18,19,20,21]. To treat PHA-based plastic waste streams, similar to complex organic waste streams, which are anaerobically digested, an open culture fermentation process could be established which exploits the use of mixed microbial cultures. With mixed microbial cultures possibly a more resilient process could be established while also other mixed plastics and/or organic waste streams could be co-fermented once the right microbial processes can be stimulated, as shown for anaerobic digestion into biogas [22,23]. If a mixed microbial bioprocess can be established using both the 3-HB and crotonate similar to the observations with the pure cultures, the carboxylates recovery from both key hydrolysates could be warranted in the envisioned PHA-based plastic recycling process. Subsequently, these carboxylates could be used as substrates for PHA production [24]. By combining all steps, a PHA cycle can be established. However, during PHA synthesis, significant amounts of CO_2_ are released. This CO_2_ could be captured and used to produce additional carboxylates and PHA using renewable electricity and microbial electrosynthesis [12].

To wrap up, so far it was not shown whether it is possible to ferment pre-hydrolyzed PHA in bioreactors with mixed microbial cultures into carboxylates such as acetate and butyrate. The functional microbial consortia as proposed could be developed to produce sole carboxylates under methane-inhibited conditions. To reduce methane production in industrial mixed-culture processes, several kinds of selection pressures can be applied [25]; for research purposes, additives such as 2-bromoethanesulfanoate can be used [26].

Therefore, the objective of this study was to identify the feasibility of hydrolyzed PHA conversion into carboxylates using mixed culture fermentation with a supply of methane inhibitor. Commercially available PHBV pellets (Tianan ENMAT^TM^ Y1000P) were used as a substrate. After mechanical and thermal alkaline pretreatments, hydrolysates collected from pretreatments achieving the highest soluble COD (chemical oxygen demand) were filtered before supplementation to anaerobic fermentation. As the inoculum, a mix of micro-organisms obtained from several carboxylates-producing anaerobic fermentations was used. The experimental results show that hydrolysates from PHA were successfully obtained and, within a couple of days, efficiently and microbially converted into carboxylates. To place this finding into perspective an outlook is provided to show how biodegradable plastic recycling via the carboxylate platform could become feasible.

## 2. Materials and Methods

### 2.1. Materials

PHBV (2% 3-hydroxyvalerate, 3-HV) pellets (Tianan ENMAT^TM^ Y1000P; Ningbo, China) were obtained from which detailed properties can be found in Appendix A. Original materials were, to avoid melting and recrystallization during grinding, mechanically pretreated by cooled milling using liquid nitrogen into ~1.0 mm powders by a miller machine (ZM1000, Retsch GmbH, Haan, Germany) with 1.0 mm mesh size [27]. Inoculums were collected from the effluents of several of our laboratory’s continuous anaerobic fermentations and/or chain elongation process bioreactors with open culture operation stored in a refrigerator at ~4 °C [28,29]. The added inoculum of 12 mL in each reactor had an initial optical density of 1.8 (and was determined at a wavelength of 680 nm (DR3900, Hach Lange GmbH, Berlin, Germany). The microbial nutrient medium compositions for the hydrolysate fermentations were the same as earlier in described batch experiments for chain elongation [30]. In addition, a methanogenesis inhibitor (2-bromoethanesulfonate, BrES) was added to prevent carboxylate degradation. Detailed information about the medium compositions is supplied in Appendix A.

### 2.2. Experimental Procedure

PHA hydrolysis was first conducted. After mechanical grinding 10 g of PHBV powders were mixed into 100 mL of total working volume (250 mL flask) and treated with acid (HCl, 0.1 M), base (KOH, 0.1 M), and demineralized water (blank). The suspensions were then put into an autoclave (CertoClav Connect, CertoClav Sterilizer, Leonding, Austria) for hydrothermal treatment (120 °C, 4 h; see temperature profile in Appendix A) [31]. After cooling to room temperature, the hydrolysates were collected and filtered with a 0.45 µm membrane (CHROMAFIL Xtra, Machinerey-Nagel, Düren, Germany). Subsequently, the pH and soluble COD (chemical oxygen demand of soluble compounds) of each sample were measured. The treated sample which showed the highest soluble COD was selected for anaerobic fermentation. All pretreatments were performed in triplicate.

The hydrolysates from the thermal alkaline pretreatment method were used as substrates for the fermentation. The final concentration of hydrolysate was 4 g COD/L, added 12% *v*/*v* inoculums, 20 mL/L stock solution, 0.5 mL/L trace metals, and 1 mL/L vitamins (Appendix A). For methanogenesis inhibition, 5 g/L of 2-bromoethanesulfonate (BrES) was added [32]. The total working volume was 100 mL with the addition of demineralized water. Next, the pH was adjusted to 7.0 ± 0.1 by 4 M KOH. After sealing the serum bottles, N_2_ flushing was used to keep an anaerobic atmosphere for 10 min. Subsequently, the headspace was filled with 1.2 bar N_2_/CO_2_ (80%/20% composition) by the gas exchanger (SC920G, KNF Neuberger, Freiburg, Germany). The microbial fermentation was performed in a temperature-controlled shaker at 35 °C and 150 rpm (see Appendix A). To limit microbial conversion, the control batch experiments were set up without inoculum, trace metals, and vitamin addition. Batch experiments were performed in triplicate. Samples were before analysis, except for pH and gas composition, collected and stored at −20 °C.

### 2.3. Analytical Methods

Soluble COD was measured with LCK514 kits (HACH GmbH, Germany) after filtration by 0.45 µm membrane and an appropriate dilution of the filtrate [28]. The total COD (TCOD) of the PHA (1588 g COD/kg PHBV on average) was measured by the COD digestion unit (C. Gerhardt Analytical Systems, Königswinter, Germany), as described in the Appendix A. The pH, gas pressure, and biochemicals were measured during two weeks of fermentation. Headspace pressure was measured using a pressure meter (GMH 3151, GHM Group, Greisinger, Regenstauf, Germany) and the gas compositions (N_2_, O_2_, CH_4_, H_2_, and CO_2_) were determined by two GC systems with standard methods at the end of the experiment [33]. N_2_, O_2_, CH_4_, and CO_2_ were measured in one GC (Shimadzu GC-2010, Kyoto, Japan) with the column in parallel, with a combination of Porabond Q (50 m × 0.53 mm × 10 µm; Varian; Part.no. CP7355, Agilent, Amstelveen, The Netherlands) and Molsieve 5A (25 m × 0.53 mm × 50 µm; Varian; Part.no. CP7538, Agilent, Amstelveen, The Netherlands). H_2_ at a pressure of 0.6 bar was used as carrier gas. The injection volume was 50 µL and the injection temperature was 120 °C. The temperatures in the oven and detector were 45 °C and 150 °C, respectively. Another GC (HP-5890, Hewlett Packard, Agilent, Santa Clara, CA, USA) with a column HP Molsieve 5A (30 m × 0.53 mm × 25 µm) was used to determine the amounts of H_2_ and CH_4_. The carrier gas was Argon, and the injection volume was 100 µL. The oven temperature was 40 °C.

Carboxylates were determined by gas chromatography (GC, Agilent 7890B, Agilent, Santa Clara, CA, USA) equipped with an HP-FFAP column (25 m × 0.32 mm × 0.50 µm). The flame-ionized detector (FID) and injection temperatures were 240 and 250 °C, respectively [34]. Samples were acidified in a final concentration of 1.5 wt% formic acids before injection, and 1 µL of the sample was injected into the column. Nitrogen was used as the carrier gas, with 1.25 mL/min for the first three minutes and 2 mL/min until the end of the run [35]. The oven temperature was 60 °C for 3 min, 21 °C/min up to 140 °C, 8 °C/min up to 150 °C and constant for 1.5 min, 120 °C/min up to 200 °C and constant for 1.25 min, and finally, 120 °C/min up to 240 °C and constant for 3 min. The 3-HB and crotonate were also qualitatively determined by GC. Chromatography data were analyzed with Chromeleon software (version 7, Thermo Fisher Scientific, Waltham, MA, USA).

## 3. Results and Discussion

### 3.1. PHBV Hydrolysis as a Result of Mechanical and Thermal Alkaline Pretreatment

To study the feasibility of carboxylate production from PHA, first, a suitable hydrolysis process should occur. The used thermal alkaline pretreatment method led to a successful PHA hydrolysis. The milled PHBV (100 g/L equivalent to 158.8 g COD/L) was estimated to be about 10% chemically hydrolyzed leading to 16.3 g COD/L of hydrolysate. The hydrolysis took place at a temperature of up to 120 °C. During this process, the pH dropped from 13.0 (0.1 M KOH) to 6.9 (Figure 2B). The whole hydrolysis process took 12 h, which led to an overall estimated hydrolysis rate of 32 g COD/L·d. The 0.1 Molar concentration was chosen to be comparable with potassium concentrations within the carboxylates-producing fermentation processes since higher concentrations may inhibit the consequent anaerobic fermentation [25,36]. The pretreatments with acids and demineralized water achieved much lower values of soluble COD (<1 g COD/L) compared to the alkaline treatment which reached more than 16 g COD/L (Figure 2A). These observations match with the results of Yu [37], whose research concluded that alkaline hydrolysis (0.1~4 M OH^−^) gained much more monomeric products than acidic solutions (0.1~4 M H^+^). In the present similar thermal alkaline treatment, both crotonate and/or 3-HB were also likely the main intermediates present in the hydrolysates [13]. In addition, minor products, likely 3-hydroxyvalerate (3-HV), 2-pentanoate, and 3-pentanoate, may be formed since they are known breakdown products from PHBV [13]. The main suspected chemicals, 3-HB and/or crotonate, were also qualitatively detected within the pretreatment (See Appendix A and Appendix A). The formation of these carboxylates does match with a decrease in the pH (from 13.0 to 6.9), as was observed (Figure 2B).

Still, 90% of PHBV polymers were not hydrolyzed. One of the reasons was that the pH (i.e., OH^−^ concentration) dropped evidently due to proton production. During hydrolysis, H^+^ is produced which consumes OH^−^ and forms water in the medium. The alkaline conditions that supported hydrolysis were not maintained. Therefore, more alkaline addition could be needed, or less PHA addition or more time could also enhance hydrolysis efficiency under non-alkaline conditions [37]. Alternatively, hydrolysis with tailored enzymes (PHA depolymerase) secreted from microbial cells could be used to improve decomposition [38]. A higher temperature may also increase hydrolysis since the increased temperature can decrease crystallinity, and subsequently enhance the degradation rate [39]. The pretreatment temperature in the autoclave was far lower than the melting temperature of the PHBV (~175–180 °C). Earlier research showed that more than 69% of the recovery of PHA depolymerization could be achieved by a thermal decomposition pyrolysis process reaching a temperature of up to 240 °C [13]. These hydrolysis products were previously used to directly produce PHA; however, this depolymerization method could also be tried to form carboxylates via fermentation. Moreover, it is well known that the particle size of the mechanical treatment can be optimized as it decides the hydrolysis speed of the PHA. Smaller particles at 100–250 µm provide a larger surface area to improve the biodegradation rate [10]. Moreover, the structure of the PHA affects hydrolysis, and Yu identified that amorphous PHB granules decomposed 30 times faster than crystallized pellets and solvent-cast films [37]. The effectiveness of the hydrolysis process is expected to be affected by the actual PHA-based product (e.g., due to additives or the presence of other polymers). Using the previously mentioned recommendations, an adapted PHA pretreatment could be developed with the increased recovery of the monomers.

### 3.2. Microbial Fermentation for Carboxylates Production

To microbially convert hydrolyzed PHA under mixed microbial culture conditions, one needs to have the right micro-organisms and operational conditions. For example, substrate and product concentrations, CO_2_ supply, pH, temperature, and used inoculum are known to affect anaerobic fermentations [25,28,29,40]. With a neutral starting pH and a diversity of inoculums, the actual alkaline pretreated PHBV was successfully microbially converted with a full conversion (measured as 106% ± 8% recovery yield) of soluble COD into acetate and butyrate (Figure 3A). Possibly also, some of the soluble COD was used for microbial growth since 3-HB or crotonate was earlier used for the synthesis of microbial biomass with pure cultures [18]. A more precise soluble COD recovery into carboxylates percentage could not be determined due to the limited accuracy of the analytical methods. The turbidity of the broth at the end of fermentation was higher than that at the start of the experiment, as observed by eyes. This supports that microbial growth occurred. The controlled bioreactor conditions produced carboxylates effectively within a couple of days (~five days) of anaerobic fermentation. The maximum acetate and butyrate production rate from day two to five were calculated to be respectively 0.54 and 0.57 g COD/L·d, which seems faster than the recent study wherein solid PHA particles were supplied to anaerobic fermentation with both biogas and carboxylates as end-products. The microbial conversion rate in this research was lower than the estimated overall pretreatment hydrolysis rate. Evidently, in the presented research, the added micro-organisms were responsible for the conversion since the control experiment without added micro-organisms (Figure 3B) did not result in carboxylate formation.

The origins of the inoculated microbes were effluents from three different anaerobic fermentation and/or microbial chain elongation reactors running on various feedstocks (including glucose, lactate, acetate, and ethanol as substrates). In these microbial communities, chain-elongating Clostridia such as *Clostridium kluyveri* are likely abundant. This species is known to have the toolset to convert crotonate in its catabolism and is possibly responsible for the formation of acetate and butyrate out of the available crotonate [20]. The 3-HB, hydrolyzed from PHB or PHBV, is possibly converted into acetate and butyrate, as is described for the species *Ilyobacter polytropus* or *Ilyobacter delafieldii* [18,41]. There are no special functional micro-organisms identified which are known to convert 3-HV into valerate besides the organisms present in the methanogenic sludge [16]. Since crotonate and/or 3-HB are convertible by a single microbial species [18,21,41,42], and 3-HV is similar to 3-HB, one can postulate that a mixed microbial process could be further developed wherein a single core microbial species is responsible for the main conversion of hydrolyzed PHA into carboxylates. Further long-term experiments and microbial-community analysis can be used to elucidate which of the functional microbial processes occur and which species are related to this. Once more biodegradable plastics are mixed during fermentation, possibly more complex microbial consortia can be established. Still, *Ilyobacter polytropus* can use both key PHB hydrolysates as well as lactate. Lactate could be obtained by hydrolyzing PLA (polylactic acids) plastics in addition to PHA [43]. Thus, mixtures of bioplastic substrates can be provided to recover carboxylates from biodegradable plastics using similar micro-organisms.

Figure 3A shows that carboxylate production was mostly complete after five days of fermentation. The concentrations of acetate and butyrate were 1.71 g/L and 1.20 g/L, respectively (Appendix A). On a molar base, one mole of acetate was formed compared to 0.48 mol of butyrate. This is comparable to the theoretical formations of acetate and butyrate from crotonate by *Clostridium kluyveri*, see Equation (1) [20]; as well as from 3-HB by *Ilyobacter polytropus*, see Equation (2) [18]). Since the amount of 3-HB and crotonate could not be revealed, a further stochiometric analysis of plausible reactions was not possible.
2.1 Crotonate + 2.2 H_2_O → 2.2 Acetate + Butyrate + 0.1 H_2_ + 1.1 H^+^     ∆G0′ = −105 kJ/mol(1)2 3-hydroxybutyrate → 2 Acetate + Butyrate + H^+^                 ∆G0′ = −59.34 kJ/mol(2)

After five days of fermentation, the concentrations of carboxylates were kept stable, which indicated that bioconversion was completed. As mentioned before, this result does support that all dissolved PHA was converted since the amount of carboxylates was similar to the added soluble COD at the start. During fermentation, the pH dropped from 7.0 to 6.1. This was similar to what was explained by Stieb’s research, where the accumulation of carboxylates (acetate and butyrate from 3-hydroxybutyrate and crotonate pure culture fermentation) resulted in a pH decrease. The earlier work controlled the pH at 7.0–7.5 to keep more stable fermentation conditions [18]. The pH in the mixed culture fermentation resulted in a pH of 6.1 (Figure 3A), while the pH in the blank groups decreased from 7.0 to 6.6, and then stayed stable (Figure 3B). This observation could be the result of some residual oligomers that might be present in the filtered hydrolysates. The oligomers may be further decomposed at the applied 35 °C temperature, which will cause further pH decrease since more monomers are produced.

During the fermentation, small amounts of caproate (33.8 mg/L) were observed too. This caproate originated from the inoculums as no increase of caproate was observed during the PHA fermentation. This relatively low concentration of caproate did not prevent acetate and butyrate formation. Earlier work showed that a caproate concentration of >20 g/L can likely inhibit anaerobic microbial conversion such as syntrophic ethanol oxidation through hydrogenotrophic methanogenesis [25]. The gas pressure maintained stability from 1.22 ± 0.03 bar at the start of fermentation to 1.23 ± 0.03 bar on the fifth day (Appendix A). Gas composition (Appendix A) measurement showed that N_2_ and CO_2_ were the two main gases and there was no methane detected in the batch bottles. CO_2_ was possibly consumed by microorganisms as CO_2_ concentrations declined from 20% to ~13%. In addition, perhaps some CO_2_ dissolved in the solution can contribute to pH decline. Trace amounts of H_2_ were sometimes detected which might be produced by microbial metabolism (H_2_ was found in two of the triplicate groups). Taking the reported reaction stoichiometry of *Clostridium kluyveri* (see Equation (1)) as an example, this does indicate that some crotonate fermentation took place as hydrogen was formed according to the observed stoichiometry.

### 3.3. Outlook–Making Biodegradable Plastic Recycling Feasible via the Carboxylate Platform

The earlier mentioned carboxylate platform could be used to recycle biodegradable plastics in the future [44,45]. The proposed route to recycle PHA to PHA again (see Figure 1) could become a useful value chain with a potential PHA recovery percentage in the range of 58–65%. This estimate was calculated based on optimized hydrolysis reaching 100%, the apparent full conversion of the produced crotonate/3HB into acetate and butyrate, thereby taking into account the microbial biomass growth yield, and earlier reported typical PHA yields from the VFA substrates [8] (see Appendix A [46]). The present study’s overall yields of the combined processes were calculated as 0.07 g acetate/g PHA and 0.049 g butyrate/g PHA (see Appendix A).

In addition to the typically mentioned biomass waste streams, other biodegradable materials such as plastics can become important input streams. Compared to the crotonate/3-HB-based recycling route [14], recycling via fermentation to carboxylates has the advantage of a wide range of other mixed- or composite-plastics and/or biomass waste streams that can be co-digested to produce carboxylates. Crotonate and 3-HB are highly valuable chemicals too and were trying to be recovered from the PHA by the formation of, for example, methyl crotonate [47]. This may be especially applicable once PHA-based products are sorted out from mixed waste streams by waste management companies. Though in case PHA is part of the composite materials, it may be more complicated to chemically upcycle PHA since the other chemicals that could thermolyze also may interfere with such recovery processes. Still, once PHAs are not sorted out of the waste stream, a co-fermentation of PHA with the other biodegradable parts present in the waste streams may become attractive [24]. Starch-based plastics and PLA packaging, for example, are also promising materials for producing carboxylates since carboxylates are likely produced as an intermediate in the anaerobic digestion of bioplastics to methane [22]. Compared to the use of biogas instead of carboxylates to remake PHA, more carbon is theoretically recovered via carboxylates than via biogas. This is because, during biogas production, CO_2_ is produced once, for example, acetate is converted into biogas [48].

Several carboxylates-producing companies, such as ChainCraft (Amsterdam, The Netherlands) [49] and AFYREN (Saint-Avold, France), could use such biodegradable plastics as a co-substrate for, possibly, carboxylates’ technical-grade applications [50]. Carboxylates could be used for applications such as feed additives, carbon sources for wastewater treatment plants, etc. [51]. While companies working on the production of PHA from biowaste streams, such as Full Cycle Bioplastics (San Jose, CA, USA) [52] and Paques Biomaterials (Balk, The Netherlands) [53], can adapt the proposed or use similar processes to close the PHA cycles via the carboxylate platform. Still, biogas from PHA is also a relevant route to recycle biodegradable plastics since companies such as Mango Materials (Oakland, CA, USA) [54] use methane as feedstock. Incorporating knowledge of bioplastic fermentation will allow companies and municipalities to improve their waste management schemes.

## 4. Conclusions

This study identified the feasibility of carboxylates production from PHA material via combined mechanical and thermal alkaline pretreatment with subsequently mixed culture fermentation. The obtained products could theoretically be used as feedstock to establish circular PHA production. Thermal alkaline pretreatment achieved an estimated 10% yield from PHBV to intermediates (attributed to crotonate and/or 3-HB). The mixed culture fermentation from hydrolysates to carboxylates such as acetate and butyrate achieved an apparent complete conversion within five days at rates of 0.8 g COD/L·d. Further insights into the working mechanisms and technical developments of microbial bioplastic recycling processes are required. The potential efficiency to recycle PHA, according to the proposed processes, was estimated to be ~60%. Furthermore, understanding how recycling routes can be implemented in bioplastic production and waste management systems is needed to establish prospective environmental–societal impact.

## Figures and Tables

**Figure 1 materials-16-02693-f001:**
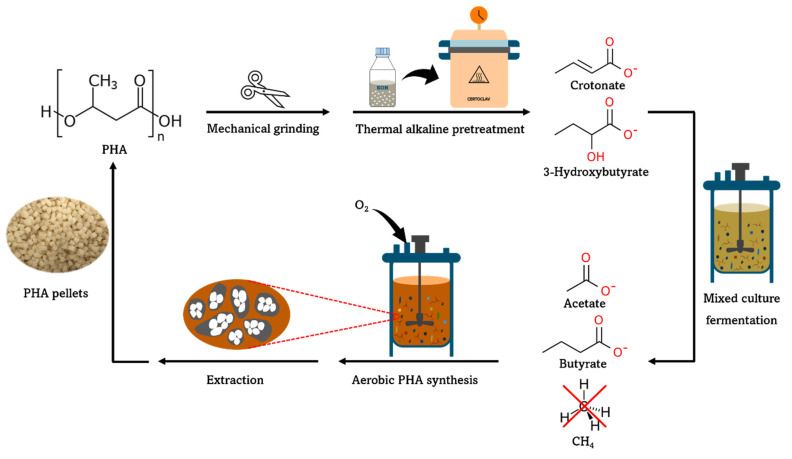
Proposed new route for recycling PHA via the formation of carboxylates after PHA application.

**Figure 2 materials-16-02693-f002:**
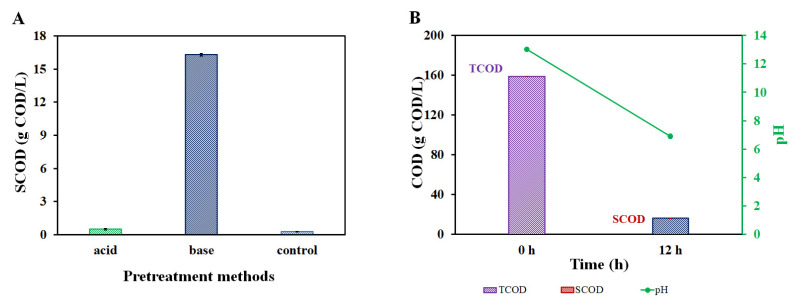
(**A**). Comparison of soluble COD (SCOD) after three pretreatments for PHBV. (**B**). COD (TCOD is the total COD of PHBV, and SCOD is measured soluble COD); and pH change between the start and end of alkaline pretreatment.

**Figure 3 materials-16-02693-f003:**
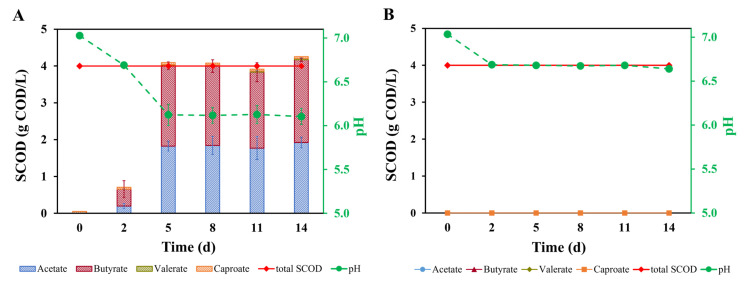
(**A**). Total SCOD of supplied PHBV (as hydrolysate), SCOD (as carboxylates), and pH during mixed-culture fermentation, (**B**). Total supplied SCOD (as hydrolysate) and pH change in the control group experiment.

## Data Availability

The data original data presented in this paper is available at the 4TU Research Database via this site: https://doi.org/10.4121/21904716 (accessed on 27 March 2023).

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
