# Peer review of "Microbial Recycling of Bioplastics via Mixed-Culture Fermentation of Hydrolyzed Polyhydroxyalkanoates into Carboxylates"

_materials, 2023, doi:10.3390/ma16072693_

Round 1

Reviewer 1 Report

The authors present recycling of polyhydroxyalkanoates way through hydrolyzation, and produced aceetate and butyrate by mixed culture fermentation. The concept is quite interesting , and it is worth considering from the perspective of recylcing process.

This manuscript would be recommended to publish after consideration as below.

1. how about value of crotonate or 3-hydroxybutyrate compared with acetate or butyrate? Also, what about use as a direct precursor for PHA production?

2. Can you estimate what percentage of PHA (weight basis) will be reproduced on a PHA based plastic waste through this process?

Reviewer 2 Report

The paper Microbial recycling of bioplastics via mixed culture fermentation of hydrolyzed polyhydroxyalkanoates into carboxylates is a well written work about chemical recycling of PHA. The topic is highly interesting for readers and for nowadays research.

The experimental work is well described and commented. However, it can be fundamentl to add analitical results showing the identification of compounds.

Other suggestions are: at the end of introductory part only the strategy and activity should be presented, not the final results. The final results should be explained briefly at the end of abstract and more extensively in conclusion

Moreover, in line 169 "dropped" should be replaced with "drop". It is suggested a control for minor spelling errors.

Reviewer 3 Report

The study is clearly presented and the experimental part appears sound and reliable The authors provide a comprehensive introduction where they give a brief state-of-the-art and a clear motivation for their study accompanied with relevant literature sources. The experimental part is also well described, the results interpretation is clear and compared to already published work backed by relevant literature sources. Personally, I like that the experimental part includes an outlook that describes how the results of this study could potentially be industrially exploited. The manuscript concludes with a short summary of the main findings as well as an outlook of still open research questions related to this study. I think this manuscript is already of high quality and I have no specific comments regarding modifications or improvements.

Author Response

We thank the reviewer for these supporting comments. there is no attachment. 

Reviewer 4 Report

General comments:

- The manuscript writeups need to be improved for more coherence and clarity.

- What distinguishes this study from others of its kind? (i.e. the pysico-chemical hydrolysis or depolymerization of PHAs was already established, likewise, there are lots of studies that already discuss PHA degradation by aerobic or anaerobic microorganisms).

- The experiment's design is not well understood (for instance, the use of liquid nitrogen to reduce sample size and non-standard inoculum collection and inoculum OD measurements, substrate concentration (COD level), COD measurement, etc.).

- Prior to its microbial bioconversion, the pretreatment of PHBV must be optimized in order to maximize the yield of carboxylates or other biochemicals during fermentation.

- For a deeper understanding of the system, microbial community analyses could be conducted both before and after the fermentation.

Specific comments:

- lines 17-18, page 1: Improve statement. Much better if yield (e.g. g acetate/g PHA) should be reported.

- line 121, page 3: Why is there a need for liquid nitrogen considering that this chemical is expensive and difficult to handle? Any more suitable options?

- line 135, page 4: What range of bases or acids were used throughout the experiment?

- lines 149-150, page 4: How was the pressure maintained and analyzed?

- lines 161-163, page 4: Why measure COD if it is not accurate to use? What is the percent accuracy of the proposed method?

- lines 156-166, page 4: Write briefly the details in the analysis of gas, PHVB, and carboxylates using GC (e.g. GC model, operating conditions, column profiles, sample preparation, sampling volume, etc.).

- lines 171-172, page 4: Is the 10% hydrolysis during base addition at 120C? What happened to the remaining 90% unhydrolyzed PHBV?

- lines 173-176, page 4: What do you mean by a time course? Kindly reframe.

- line 177, page 5: Use demineralized or deionized water instead of demi-water?

- lines 184-187, page 5: Probably, only a few were hydrolyzed to 3-HB or crotonate using the proposed procedure. Hence, there is no change in pH.

- Figure 3:Aside from the COD measurements, include the carboxylate concentrations as determined by the GC.

- lines 241-252, page 5: I suggest investigating the microbial community of the system.

Round 2

Reviewer 4 Report

Please resubmit the manuscript after extensive English editing has been done.

Author Response

Dear reviewer,

We earlier accidentally uploaded the wrong file. Instead of uploading the revised manuscript we apparently uploaded the supplementary information.  Find hereby again the response to your review report as well as the revised manuscript with tracked changes.

Kind regards.

Round 3

Reviewer 4 Report

The manuscript has improved, just a few minor comments below:

- line 128, page 3: Why an OD of 1.8 was selected? Is it the final OD of the culture? Kindly explain.

- Figure 1, line 100, page 3: Remove the statement.

- lines 162-163, page 4. Indicate the model of the used GC columns and their corresponding operating conditions (oven/injection/detection temp., carrier gases, etc.). 

- lines 182-183, page 5: Indicate the starting/initial COD of the sample PHBV material.

- Figure 2, lines 200-201, page 5: Remove the statement.

- lines 242-244, page 6: Kindly reframe the statement.

- Figure 3, page 6: Revise the figure title. What do you mean by "total supplied SCOD"?

- Figure 3, lines 249-250, page 6: Remove the statement.

Author Response

Dear Reviewer,

Thank you for your suggestions and questions!

We updated the manuscript accordingly and provided the rebuttal and a revised manuscript with all tracked changes.

Best regards, 

Yong Jin, Kasper de Leeuw & David Strik.
